# First Encounters with a Robot: The Value of Augmented Reality for Non-Experts Performing Multiple Tasks with a Mobile Robot

Andre Cleaver
andre.cleaver@tufts.edu
Tufts University
Medford, Massachusetts, USA

Victoria Chen
victoria.chen@tufts.edu
Tufts University
Medford, Massachusetts, USA

Jivko Sinapov
jivko.sinapov@tufts.edu
Tufts University
Medford, Massachusetts, USA

## ABSTRACT

Augmented reality (AR) devices enable robots to convey their sensory and internal cognitive representations (e.g., 3D points clouds, motion plans, etc.) to humans using graphical images that are rendered in context of their environment. AR devices for robot training and education introduces users to a new perspective in robot interaction. For instance, users can validate or debug a robot's internal representations and environmental perceptions as well as explore the robot's internal state and capabilities. Although AR devices in previous human-robot interaction studies have assisted users in single robotic tasks, it is unclear in what settings users will want to utilize an AR device across multiple types of interactions to accomplish their tasks. Specifically this paper aims to understand how users use AR, what value it brings, and in which types of robotic-related tasks it is most useful. We conducted a human-participant study where 19 participants completed 5 unique robotic activities. We found that users in our Test group with access to an AR device had greater performance compared to the Control group that did not have an AR device and preferred using an AR device in activities that relate to Object Detection and Safety.

## KEYWORDS

Augmented-Reality, Data-Visualization, Human-Robot Interaction, Robotics

**ACM Reference Format:**
Andre Cleaver, Victoria Chen, and Jivko Sinapov. 2023. First Encounters with a Robot: The Value of Augmented Reality for Non-Experts Performing Multiple Tasks with a Mobile Robot. In *Proceedings of Companion of the 2023 ACM/IEEE International Conference on Human-Robot Interaction (HRI '23 Companion).* ACM, Stockholm, SE, 7 pages. https://doi.org/XXXXXXX.XXXXXXX

## 1 INTRODUCTION

People working with a robot may find it helpful to be aware of the robot's internal representation of its environment (i.e., laser

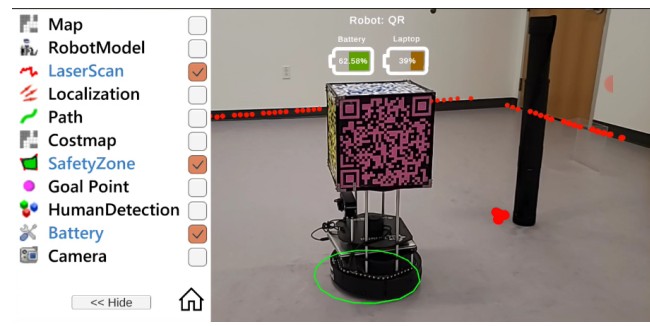

**Figure 1: Screenshot of *SENSAR* on a tablet. The application detects the robot's pose with the target image cube and renders the selected robotic data types in relation to the cube.**

scan data) for collaborative task completion, debugging, and educational purposes. Roboticists traditionally use software like *RViz*[1] to visualize the robot's sensory and cognitive data. A drawback of RViz is that users have to perform mental transformations on the visualized data due to the frame misalignment from the screen to the environment. To alleviate this drawback, Augmented Reality (AR) devices have been integrated with robotic systems which now can render the data over the environment [4–7, 11], ultimately minimizing any ambiguities that might arise with misaligned reference frames. However, there is a need to understand how users use AR, what value it brings, and in which types of robotic-related tasks it is most useful. In this paper, we address this need in 2 parts: 1) we developed a mobile AR application, *SENSAR*, that rendered robotic data in context of the environment (see Figure 1), and 2) we conducted a human-user study where participants completed robotic activities that differ in types of robotic information: *Sensors*, *Cognitive-Decisions*, *Diagnostics*, *Safety*, and *Actions*. We measured performance metrics when comparing a Control group that used only RViz with a Test group that has access to both *RViz* and *SENSAR*. We further analyzed the Test group for user preference between visualization tools.

We hypothesized that users will complete tasks with better performance with the use of AR, and that they will prefer using an AR device more in tasks that relate to *Sensors*, *Cognitive Decision*, *Safety*, and *Actions*. Our results show that users found value in using AR in tasks that involve user movement within the environment to further obtain the spatially rendered data such as the *Sensor* and *Safety* tasks. Participants whose utilized the AR device in *Sensors*

---
[1]http://wiki.ros.org/rviz

and *Safety* activities showed an increase in task performance compared to the Control group. Contributions in this work include our open-source AR system for Human-Robot Interaction as well as insights into the uses of AR devices in robotic training and learning.

## 2 RELATED WORK

Research groups often develop AR robotic systems for a single type of interaction with a robot (i.e., reprogramming, collaborative task). Chong *et al.* [3], Rosen *et al.* [17], and Ong *et al.* [13] introduced an AR reprogramming system for industrial arm manipulators allowing users to guide and plan the robot's path trajectory. Gruenefeld *et al.* [9] and Chan *et al.* [1] also used an arm manipulator to evaluate their AR safety system in collision avoidance tasks. Walker *et al.* [20] created an AR motion intent system that conveyed an aerial drone's future path trajectory in an assembly task, while Newbury *et al.* [12] visualized a robot's intent during object handovers. Williams *et al.* [22] augmented a robot's deictic gestures for guided attention tasks. Weber *et al.* [21] developed an AR calibration system to account for sensor misalignment on a robot. Xiang *et al.* [23] demonstrated a projection-based AR safety system with a mobile robot to visualize construction information due to potential health and safety concerns with using an AR head mount display on a construction site. Unlike the mentioned studies which focused on a single task, we evaluated participants in multiple types of interactions with a robot and determined if they found value in using AR devices.

We developed our *SENSAR* application to visualize multiple types of robotic data for multiple types of interaction with a mobile robot.

AR devices have also been integrated for educational robotic purposes (i.e., debugging, programming) often as part of research group studies. Pasalidou *et al.* [14] aimed to create an immersive environment to increase student engagement in a robot programming activity. Radu *et al.* [16] reported the impact of unequal access to AR in group collaborative tasks with a robot in programming activities. Cheli *et al.* [2] observed students' behavior toward using the AR device while debugging their robot after unexpected behaviors. Villanueva *et al.* [19] created a robotic tele-consulting system for distance learning for students and instructors through a shared AR makerspace. Participants in these studies are often in a position where it is required to use the assigned AR device to complete their task. In our work, we evaluated participants' performance and preference in completing multiple types of robotic tasks when provided the option to use either *SENSAR* or RViz.

## 3 METHODOLOGY

We designed a *2 x 5* within and between human-user experiment. For between-users, each user was randomly assigned to either the Control or the Test group. The Control group had access to only the computer-based RViz, the traditional robotic data visualization tool. The Test group had access to both RViz as well as our Augmented Reality (AR)-based version of RViz, *SENSAR*. Both visualization tools streamed the same data in terms of size, shape, color, and quantity (see Figure 2); therefore, the primary difference between the tools is data placement. For within-users, users completed all 5 robotic activities that each represented the categories of robotic

information: *Sensors*, *Cognitive-Decision*, *Diagnostics*, *Safety*, and *Actions*.

### 3.1 Hardware and Software

Users interacted with a *Turtlebot2* robot controlled with *Robot Operating System* (ROS) [15] running on Ubuntu 16.04. The robot was equipped with an *RPLIDAR A2M8 360°Laser Scanner*. The robot's sensors and internal data were extracted and filtered from ROS using Python and C++ scripts. Participants in the Control and Test group had access to a desktop computer with a standard monitor screen running RViz. Participants in the Test group additionally had access to an AR device (Samsung Galaxy tablet S8) running *SENSAR*, an *Unity*[2] application that rendered the robot's data in respect to the physical robot. *SENSAR* followed a similar approach to visualizing data through mobile devices as in Zea *et. al* [24].

*Vuforia*[3] tracked the robot using a target-image cube fixed to the robot (see Figure 1). Data exchange between the robot and AR device occurred over a shared Wi-fi network using *ROS-Sharp*[4]. The *SENSAR* project can be found here, *SENSAR_UNITY*[5], *SENSAR_ROS*[6].

### 3.2 Users and Procedures

We recruited 19 participants (11 males, 7 females, and 1 who did not answer) around the University Campus. Participants' age ranged from 19 to 36 ($\mu$=23.5, $\sigma$=4.4). Participants reported on a 7-point scale their familiarity with robots ($\mu$=4.05, $\sigma$=2.5), visualization tools ($\mu$=3.3, $\sigma$=2.2), and augmented reality (AR) ($\mu$=1.85, $\sigma$=1.03). Here, the higher number means more familiarity and experience. 12 participants were selected for the Test group and 7 were selected for the Control group.

The study was conducted in a controlled lab setting as shown in Figure 3. Participants first provided consent to participate in the study after reading the study description followed with a pre-questionnaire that gauged their experience with robots and AR. The lab conductor then provided a short tutorial on how to use the visualization tools, RViz and the AR device, with respect to the selected condition. Participants then interacted with the robot and visual tools for a few minutes to familiarize themselves with the technology. A reference sheet[7] was provided which showed the location of useful features in RViz such as changing screen views, toggling visualized data sets, and mouse Control functions. Once a participant was ready to proceed, the study continued with completing the 5 robotic activities that followed the same order for each participant. Each session took approximately 60 minutes to complete the entire study.

### 3.3 Robotic Activities

We created the following robotic visual categories: *Sensors*, *Cognitive-Decisions*, *Diagnostics*, *Safety*, and Actions after considering the taxonomy discussed by Hedayati *et al.* [10] on the types of information robots might signal to humans. We reasoned that Privacy

---

[2]https://unity.com/
[3]https://developer.vuforia.com/
[4]https://github.com/siemens/ros-sharp
[5]https://github.com/DreVinciCode/SENSAR_UNITY
[6]https://github.com/DreVinciCode/SENSAR_ROS
[7][Reference Sheet]

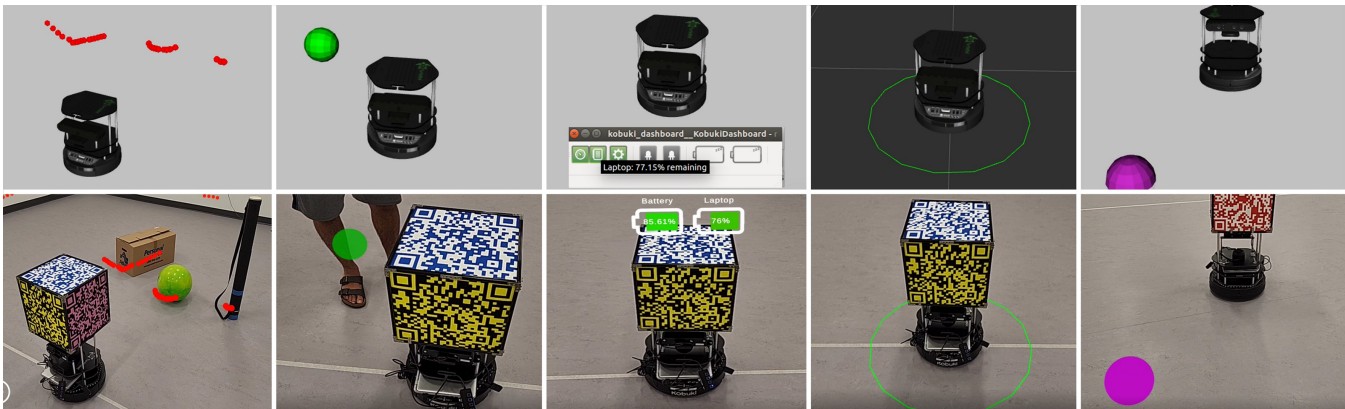

**Figure 2: Visualizing different robotic data set types shown in RViz (top) and through an AR device (bottom). From left to right, Laserscan (Sensors), Person Detector (Cognitive), Battery Levels (Diagnostics), Safety-Zone (Safety), and Goal Point (Actions)**

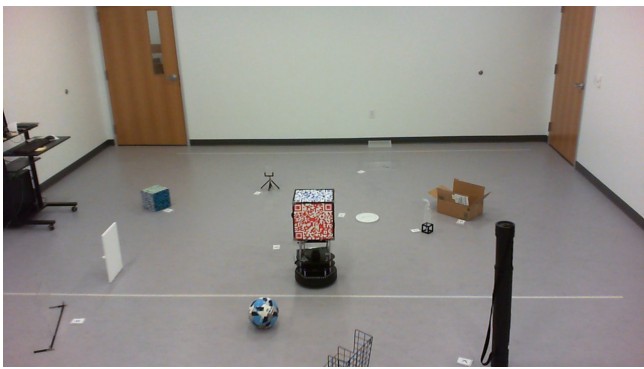

**Figure 3: Lab space where experiment took place with robot in the center. Objects in scene were part of the Object Detection task. The Control group used a desktop computer that remained to the side of the room (left side of image).**

includes any raw data that the robot is capturing, and therefore we generalized Privacy as Sensors. We reasoned that high-level information within Communication can be generalized to Cognitive Decisions. Cognitive Decisions includes any algorithms such as object-detection and navigation planning. We generalized Condition to be part of Diagnostics which includes the robot's internal states and specifications, such as battery life and payload capacities. Safety refers to requirements by the robot to operate autonomously, such as a unoccupied clearance zone. This clearance zone, the area around the robot, must be cleared by any objects in order to continue operating. Actions refers to actions taken by the robot such as navigation goals points.

We created robotic activities that we believe best represented each category. We understand that each task can be modified in terms of complexity, but for this study we purposely made each activity as simple as possible so that each task can be completed with either visualization tool. The following goal oriented activities were created:

*3.3.1    Object Detection.* In this activity, each participant must determine which objects are detected by the robot's lidar sensor. The objects remain fixed to their positions and have labels placed on the ground for participants to reference. Various objects that differ in material, size, and height placement are positioned arbitrarily within the room surrounding the robot (see Figure 3). If some items fall below the lidar sensor's plane of detection, it will not be detected. Other items were transparent and will also not be detected. Some objects were clustered together purposely to ensure ambiguity. Participants reported their answers verbally to the lab conductor. For this task, we measured total items correctly identified and how confident they felt completing the task. For the Test group, we also measured total time spent on each visualization tool.

*3.3.2    Calibration.* In this activity, each participant was instructed to calibrate the robot's *person-detection*[8] program. The robot determines the likelihood of a person through a leg-detection algorithm which utilizing the robot's lidar data as input and approximates the location of that person or multiple people. The program then assigns a reliability value for each detected "person" and will only output measurements if the reliability value is greater than a threshold value. Participants were instructed to adjust the *person-detection* program by changing this threshold value. Participants calibrated the program by adjusting a slider located on a separate laptop that corresponded to the threshold value parameter. Participants reported the threshold value that they believe best identified both the lab conductor and themselves. For this activity, we only determined if the participant was able to detect the lab conductor. For the Test group, we additionally measured total time spent on each visualization tool.

*3.3.3    Battery Level.* In this activity, participants were required to report the robot's internal battery-levels (laptop and Turtlebot base). Robotic data in this category is not supported in visualization tools. Because there are no spatial coordinates linked with data in this category, we utilized the *Turtlebot_Dashboard*[9] program that showed the robot's battery-levels in a separate computer window for the

---

[8]http://wiki.ros.org/leg_detector
[9]http://wiki.ros.org/turtlebot_dashboard

Control group. The battery-levels on the AR device were rendered on top of the robot in a similar format as seen in Figure 2 center. For this activity, we recorded whether or not the participant was able to report the battery levels. For the Test group, we measured total time spent on each visualization tool.

*3.3.4 Safety.* In this two part activity, participants were positioned in the middle of the room along a line, and the robot was positioned on the other end of the room, opposite of the user. Participants were instructed that they must reposition themselves outside of the robot's "safety clearance zone" while remaining on the line as the robot moves towards the opposite end of the room. The safety clearance zone resembled a green ring located on the ground that surrounds the robot (See Figure 1). Participants performed this task in two rounds with the ring size changing in each round. For part two, we gauged the participant's belief on the radius length of the safety ring by requesting them to mark a minimum and maximum range from the robot. Participants marked the location of the ring while the robot was stationary with a tape on the ground. Participants who are very confident in the ring's location will likely have shorter distances between the two markers as opposed to those who are unsure marking larger distances. For the Test group, we also measured total time spent on each visualization tool.

*3.3.5 Goal Point.* In this activity, participants were instructed to Control the robot via teleoperation and reposition the robot as close as possible to a goal-point. The robot begins at an assigned starting position and a single reachable goal-point was marked on the robot's internal map (both in RViz and SENSAR). The Control group teleoperated with the robot via keyboard, while the Test group could use the AR-device. For this task, we recorded whether or not the participant was able to teleoperate the robot over the Goal Point. For the Test group, we also measured total time spent on each visualization tool.

With the following study structure, we formed the following hypotheses:

> **H1**: Users with access to an AR device will complete the robotic activities more accurately than users with access to only a computer.
> **H2**: Users with access to an AR device will preference using the AR device more than the computer in robotic activities Sensors, Cognitive-Decisions, Safety, and Actions.
> **H3** Users engaged with robotic tasks with access to an AR device will spend more time on the AR device than on the computer to complete the Sensors, Cognitive-Decision, Safety, and Action activities.

## 3.4 Measures & Analysis

To answer our questions, we gathered a combination of objective and subjective measures. Video recordings with timestamps captured the interactions of the participants with the robot and visualization tools.

Objective measures for both groups include performance for the Object Detection and Safety Task. To measure performance for the Object Detection task, the average accuracy was calculated and compared between the Control and Test group. Accuracy was

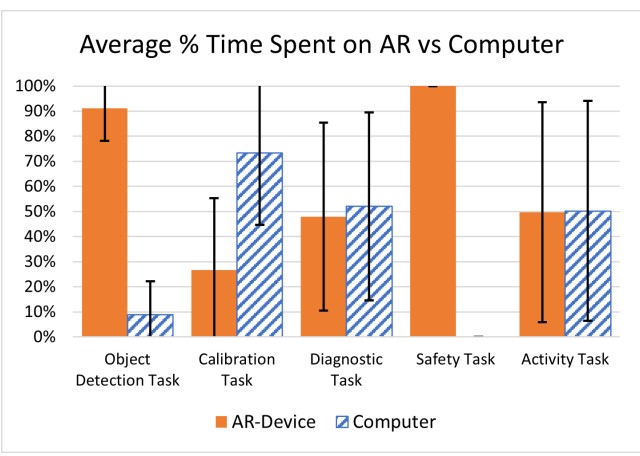

**Figure 4: Plot of average time spent on visualization tools among the Test group for each task. Solid black bars represent the standard deviations for the Computer, and dashed black bars represent standard deviations for the AR device.**

calculated by adding the total number of correctly classified objects (i.e., Detected, Not detected) and dividing by the total number of objects in the environment. For the Safety task, we compared the average marked distance between the Control and Test group. The remaining tasks only checked if the participant was able to achieve the goal. In the Test group, we measured duration spent on each visualization tool and calculated the average percentage of total time should participants decide to use one tool over the other. Subjective measures include 7-point Likert-Style ratings on user preference towards visualization tools for each task as well as a modified version of the *Questionnaire for the Evaluation of Physical Assistive Devices (QUEAD)* [18]. Open-ended responses were gathered for qualitative feedback. Questions included: "Describe your experience completing the task." In the Test group, we added "What made you use the AR device over the computer?" and "What would you like changed about the AR device?" We analyzed our data using ANOVA and post-hoc tests to determine any significant differences.

## 4 RESULTS

The average accuracy and standard deviation for the Object Detection task between the Control and Test group was ($\mu$=67%, $\sigma$=13.49%) and ($\mu$=87%, $\sigma$=7.89%) respectively. T-test showed significant differences between groups (p<0.05). The safety ring location task in which participants marked the inside and outside of the ring had an average length of ($\mu$=13.28 cm, $\sigma$=6.85 cm) and ($\mu$=4.29 cm, $\sigma$=2.42 cm) for the Control and Test group respectively. The large difference in average suggests that the Control group over estimated the radius of the circle due to their uncertainty of the ring's location. T-test showed significant differences between groups (p<0.05) All participants successfully completed the calibration, battery level, and Goal Point activity with no significant differences.

*4.0.1 H2.* Figure 5 shows the average preference rating towards the AR device for each robotic activity within the Test group. Object

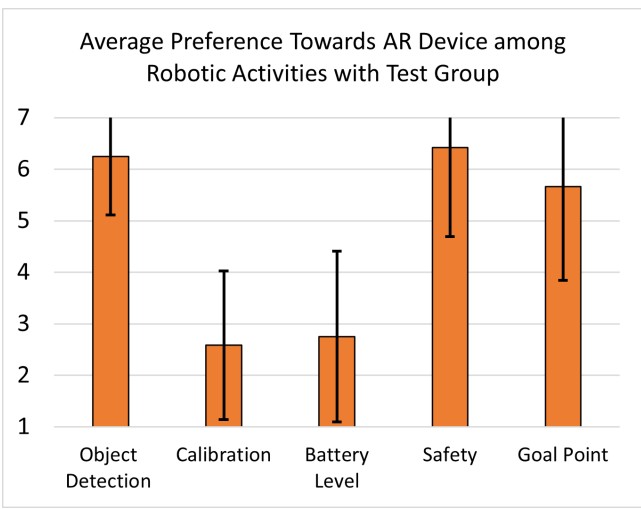

**Figure 5: Plot of average 7 point Likert-Style on user preference towards the AR device for each robotic activity within the Test group. Solid black bars represent standard deviation.**

Detection ($\mu$=6.20, $\sigma$=1.13), Safety ($\mu$=6.42, $\sigma$=1.72), and Goal Point ($\mu$=5.67, $\sigma$=1.8) averaged towards using the AR device, while Calibration ($\mu$=2.58 $\sigma$=1.44) and Battery Level ($\mu$=2.75 $\sigma$=1.65) averaged more towards using the Computer. ANOVA test showed significant differences among the activities, $F(4,55)$= 17.5, ($p$<.001). Levene's test showed no significance indicating no differences in variability within the groups, ($p$=.457).

Figure 6 shows the average ratings using the QUEAD questionnaire within the Test group. Here, the higher the rating, the stronger they agree with the statement. Standard T-tests showed that there were significant differences for several items: Good Idea ($t(12)$ = 2.53, $p$=.019, Likeness ($t(12)$ = 3.09, $p$=.005, Rigid & Flexibility ($t(12)$ = -4.33, $p$=.001, Easy to Learn ($t(12)$ = 2.64, $p$=.015, Effectiveness ($t(12)$ = 2.86, $p$=.009, Rapidness ($t(12)$ = 2.47, $p$=.022, Efficiency ($t(12)$ = 2.30, $p$=.031, Performance ($t(12)$ = 2.21, $p$=.038, and Usefulness ($t(12)$ = 2.530, $p$=.019.

*4.0.2 H3.* Figure 4 shows the average time spent on each visualization tool within the Test group for each robotic activity. Here, participants mainly used the AR device for the Object Detection task. Participants mainly used the computer for the calibration activity. Diagnostic and Goal Point activities showed roughly an equal amount of visual tool usage. The Safety activity is the only task where all participants used the AR device. T-tests revealed significant differences for the Object Detection and calibration task ($p$<.001).

## 5 DISCUSSION

**H1** stated that users with access to an AR device will complete robotic tasks with better performance than users with access to only a computer, and our results supported this claim in the Object Detection and Safety activities. The remaining activities showed no differences. Activities Object Detection and Safety required relatively precise spatial targeting/localization of the data to perform

well. AR devices reduced the ambiguity that Rviz created with the visualized data superimposed over the environment. Participant responses supported this claim:

**[Participant:7]** "*It helped you see the things you needed to see in the room instead of on a computer screen.*"

**[Participant:9]** "*It's convenient to see the where the dot actually in the real world is.*"

**[Participant:10]** "*it's easier to identify the destination using AR because you can actually stand there. You cannot achieve this using the computer.*"

In the *Action* activity however, RViz provided a top-down view of the study space, virtual model of the robot, and markers for the goal point. Nearly all participants that used RViz controlled the robot via keyboard until the virtual robot and marker aligned, all while not looking at the physical robot. Few participants reported the advantages of having a top-down view:

**[Participant:7]** "*You could basically see everything all at once instead of having to go in and out of menus*"

**[Participant:11]** "*having the overhead perspective in some cases made things easier to pinpoint, like the goal point exercise*"

**[Participant:12]** "*it's more straightforward to use the top/perspective view to capture the entire frame, and don't really need to consider obstacles in the space.*"

**H2** stated that users with access to an AR device will prefer using the AR device more than the computer, and our results supported this claim for the Object Detection, Safety, and Goal Point activities. Calibration and Diagnostic information can potentially be visualized on a computer window rather than a panel in world space; therefore, users may not bother using an AR device to retrieve such information as reported by one user:

**[Participant:9]** "*For display of data not related to locations (like battery levels), it's easier if we could just see it in the computer (so we don't need to hold a tablet when checking battery levels*"

In addition, the AR device yielded better accuracy compared to the computer in almost all items in the *QUEAD* survey. When working with AR devices in the context of robotic activities, it is worth to consider the AR design interface for each task to improve user experience as one user suggested:

**[Participant:7]** "*Somehow make it so you can see all of the information without having to point the camera at the robot.*"

Although our tablet was capable of running our AR application, smooth tracking of the robot while in motion proved challenging. In our case, once the target image cube moved out of view of the camera, the visualized robot data was often rendered out of frame. To prevent any confusion, all virtual images were rendered only if the target image was in view of the AR device. In addition, participants in the Test group were required to hold the tablet and have the camera focused on the robot to see the visualizations. This requirement may not produce the most comfortable or natural interaction with the robot especially for any long term interactions times, and we noted that some participants leaned towards using the computer rather than the tablet to avoid holding the device. It may be important to shift towards using head-mounted augmented reality devices rather than mobile AR devices to improve user experience.

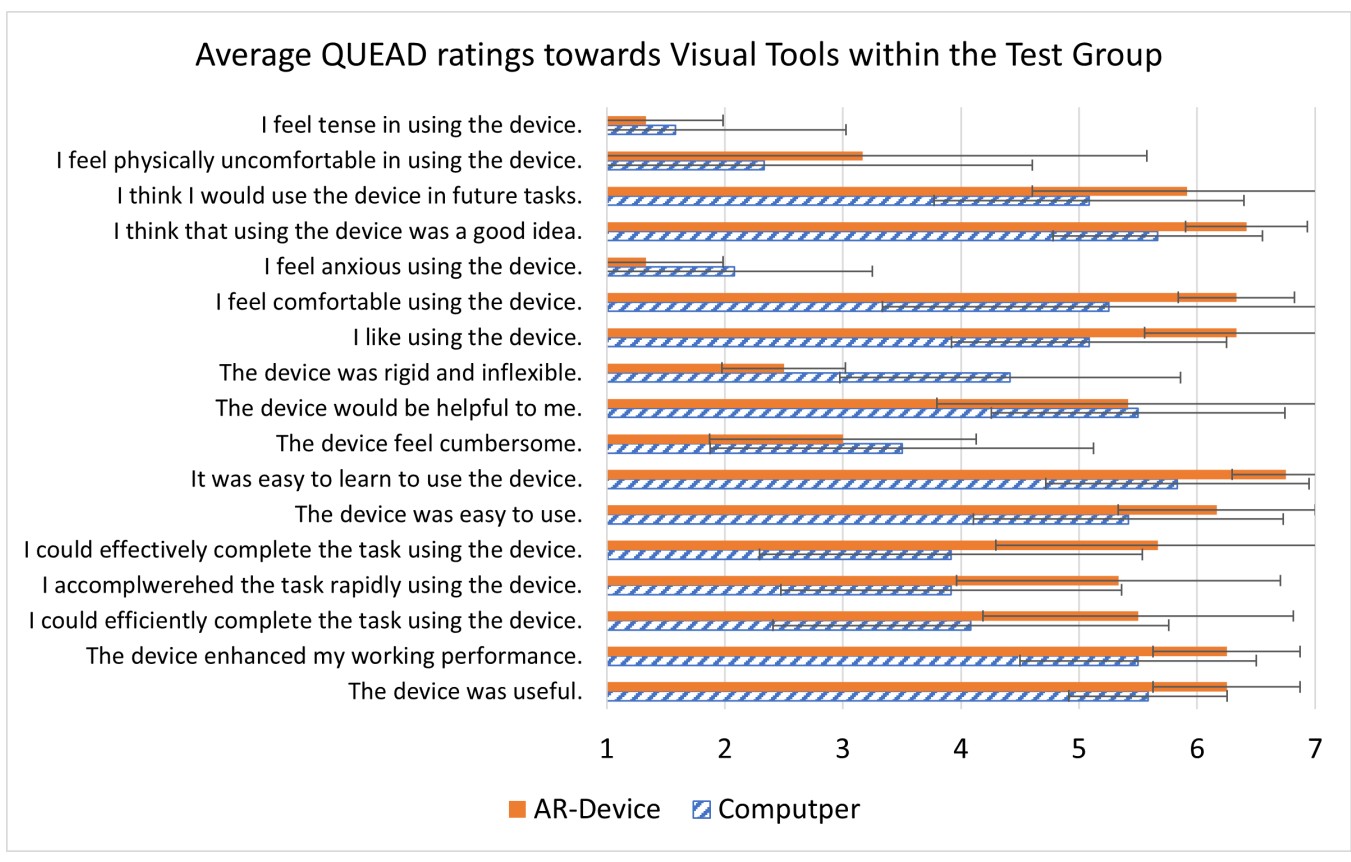

**Figure 6: Plot of average QUAED ratings within the Test group. Asterisks (*) indicate significant differences found. Solid black bars represent standard deviation.**

**H3** stated that users will spend more time using the AR device than on the computer to complete the robotic activities, and our results supported this claim for the Object Detection and Safety activities. It is worth considering the actions of the human when deciding if an AR device is a viable visualization tool for a robotic activity. The Safety activity was the only task where participants were instructed to physically move in the environment with the robot in motion, and all participants used the AR device the entire time to accomplish that activity. During the Object Detection and Goal Point activities, most users took advantage of moving within the environment with the AR device to better comprehend the visualized data. Therefore, we suggest AR devices as a valuable visualization tool for highly active activities.

## 6 CONCLUSIONS AND FUTURE WORK

In this paper, we conducted a human-participant study to determine which types of robotic data do users visualize with augmented reality (AR) when completing tasks with a ground mobile robot. Participants were assigned tasks that differ in the type of robotic information needed to successfully complete the tasks. The types of robotic data included: *Sensors*, *Cognitive-Decisions*, *Diagnostics*, *Safety*, and *Actions*. Participants in our Test group additionally had

*SENSAR*(AR) that visualized the mobile robot's sensory and cognitive data. Our human-user study revealed that robotic activities *Sensors* and *Safety* benefit greatly with AR devices with results showing greater performance, preferences, and rating towards AR devices. We found little differences in *Diagnostics* and *Calibration* between performance and preferences in computer and AR groups

While our study highlighted the value an AR device can have for users completing robotic activities, there are few limitations. The number of recruited participants in total as well as the size of each condition may not be adequate for our statistical analysis. Further recruitment may solve this issue. We understand that we are tackling a broad question, but the results from this study helped point out our next direction. We aim to dive further into the Sensors and Safety data categories and consider more complex tasks. We believe these tasks will reflect those performed by factory workers that encounter robots frequently. We also understand that the object detection task only rendered the robot's Lidar sensor data for both conditions. The physical objects used for detection were not rendered in the virtual environment. We argue that the robot's only output was the Lidar readings, and visualising representations of the physical objects is added information.

Another limitation is our AR application relied on a mobile tablet and target-images to track the robot which as a result required users

to have the robot in view of the AR device to render any robotic information. The computer running RViz remained in one location of the room, and few users questioned whether they can move the location of the computer.

A direction worth exploring is the visuals used to represent robotic information. Although we aimed to keep the visual representations consistent between both visual tools, participants may benefit with more clear and visually appealing designs as discussed in Groechel et al. [8].

For future work, we want to explore how users utilize AR devices in first encounters with a robotic arm manipulator compared to a ground mobile robot when completing similar robotic activities. Reinforcement learning (RL) in the context of robotics has remained as a black box for most humans and therefore, we also plan to evaluate an extension of *SENSAR* [25] designed for RL and determine if it enhances human-in-the-loop RL.

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
