# OpenReview forum: "First Encounters with a Robot: The Value of Augmented Reality for Non-Experts Performing Multiple Tasks with a Mobile Robot"
_humanrobotinteraction.org/HRI/2023/Workshop/VAM-HRI — VAM-HRI 2023 Oral_

### Official Review · Program_Chairs · 2023-02-24
**Accept**

**Rating:** 8
**Confidence:** 5

**Review:**

Reviewer 1: This paper discusses the implications of using a desktop interface (rviz) vs an AR interface (SENSAR) + rviz across a multitude of tasks (5 total) validating via a user study (between + within, n=19). They found users generally preferred AR over desktop finding a few differences between different tasks.


I recommend accepting this paper to the workshop. It should bring about good discussion regarding use cases of AR across a multitude of tasks highlighting different tasks have different benefits. I think this work has an exciting set of new directions and look forward to it at the workshop!


Fined-grained comments:
1. Many of these are personal preferences or nitpicks, the work is good so adding anything I came across as potential comments
2. I would personally change the title of this paper. Learning about Mobile Robots implies this is an educational paper (i.e., using AR robotics in a classroom setting); possibly using the words “non-expert/new user” or referring to the “user” in the title, just an idea; Further, I think the strength of the paper is that it has multiple tasks with slightly different lessons for each so I would include some reference to “multiple tasks” in there
3. Put the number of participants in the abstract (i.e., $n = 19$)
4. Please also indicate the results between control and test in the abstract, whether they are non results or “good” results
5. You have a leftover “VC” commend in paragraph 2 in Introduction
6. Possibly relevant background papers, not saying the must be added but worth looking into in my opinion:
   1. [AR rviz] AR Indicators for Visually Debugging Robots https://dl.acm.org/doi/abs/10.5555/3523760.3523958
   2. [AR rviz] iviz: A ROS visualization app for mobile devices https://www.sciencedirect.com/science/article/pii/S2665963821000051
   3. [AR rviz] Reimagining RViz: Multidimensional Augmented Reality Robot Signal Design https://ieeexplore.ieee.org/document/9900692
   4. [multitasking AR + robot] Design and Evaluation of an Augmented Reality Head-mounted Display Interface for Human Robot Teams Collaborating in Physically Shared Manufacturing Tasks https://dl.acm.org/doi/full/10.1145/3524082
7. Super nitpicky: “We understand that RViz has an advantage over our SENSAR system such as multiple viewing options and camera zooming” -> I think this is better framed as a possible benefit. The two platforms (desktop and mobile) have been tested against each other before in VAM and I think it is more about mentioning there are benefits to both and highlighting the differences (the goal isn’t to say AR is always better but to highlight when/why it would be better and this sentence makes it seem a bit more like you want AR to be better
8. H1: I would avoid the word “better” and use a more exact word (e.g., “faster/more accurately”); this applies to any time you refer to “performance” as this can be measured in a variety of ways, primarily accuracy, speed, and precision
9. H1L Also, I would remove the reasoning behind each within the hypothesis. You can speculate on the reasons and back them up but putting them inside the hypothesis can conflate results with the reason behind the change. You tested AR vs desktop and the way you can say “ AR devices can render data directly over the environment thus minimizing any ambiguities that may arise” would require user interviews who used both AR and desktop and them citing this as something they preferred;
10. The above H1 applies to H2 and 3 as well
11. Good reporting on recruitment and participant information although I am finding it difficult to see the breakdown number of participants in the Control via Test groups
12. Inconsistent capitalization of “Control”
13. “The only difference between the tools is data placement” -> I would say “primary” as there are inherently different interaction paradigms between the two (e.g., a kinesthetic vs static interaction, also you can technically/theoretically stream an external camera/depth feed into rviz and place data in a 3D depth via/reconstruction).
14. I’m a little wary of t-tests given the population size (also not sure the Test v Control sizes); I would personally acknowledge this as a limitation of the work.
15. Figure 4 is difficult to parse. It says average time but then uses percentages as a scale. Possibly you mean Average % of time using AR vs computer. I also prefer traditional, non-stacked bar charts as they are much easier to compare (https://www.fusioncharts.com/blog/a-fresh-look-at-stacked-bar-charts-the-worst-or-the-best/)
16. I appreciate the use of the blue checker pattern under the assumption that this is for people who are color blind or viewing in black and white. To keep this, please change the pattern to something like a striped bar (https://i.stack.imgur.com/ZhTlA.png) with an outline around the whole bar (most important bit being the outline around the bar as it is difficult to see/compare the ends such as in Fig. 6)
17. Make sure to be clear that the test group has AR + rviz (see paragraph 1 of Section 6), add this as “additionally had SENSAR/AR”, a diagram for the user study itself may be helpful to delineate the breakdown of conditions/sizes of groups

Reviewer 2:

This paper evaluates how effective AR is for a variety of human-robot interaction tasks to better understand when it is more useful compared to traditional interfaces (e.g: RViz). Overall, I recommend accepting this paper to the workshop, it is relevant to the community and offers useful insights into better understanding in what HRI settings VAM technologies are most effective.

Feedback:
- The goal point task is unique in that it involves teleoperation, which introduces two components: control and visualization, whereas the other tasks don’t involve controlling the robot. Therefore, there needs to be a separate evaluation of each visualization and control method to understand what is having the more important impact (the difference in control, or the difference in visualization). For example, [1] found that position tracked controllers for VR teleoperation are more important than the 3D HMD for visualization.
- While I understand that the robot only had access to a 1D laser scan in this case, I wonder how significant the differences in performance in these tasks would be if the user saw a 3D colored point cloud in RViz, similar to [1], especially since it was mentioned users wanted to walk around the space. An evaluation of how sensor modality impacts the usefulness of AR devices (in conjunction with task) in HRI settings would be interesting.
- In the introduction first page, there is a comment [VC: ] that should be removed.

[1] Whitney, David, et al. "Comparing robot grasping teleoperation across desktop and virtual reality with ROS reality." Robotics Research: The 18th International Symposium ISRR. Cham: Springer International Publishing, 2019.

---

### Decision · Program_Chairs · 2023-03-02

Accept (Oral)